# Transcriptomic Responses of Blue Bat Star *Patiria pectinifera* to Sediment Burial

**DOI:** 10.3390/ijms26115208

**Published:** 2025-05-28

**Authors:** Han Dong, Linli Wan, Chunsheng Wang, Cong Sun, Xiaogu Wang, Lin Xu

**Affiliations:** 1College of Life Sciences and Medicine, Zhejiang Sci-Tech University, Hangzhou 310018, China; 202220901009@mails.zstu.edu.cn (H.D.); 2023210901076@mails.zstu.edu.cn (L.W.); michael_sc@zstu.edu.cn (C.S.); 2Key Laboratory of Marine Ecosystem Dynamics, Second Institute of Oceanography, Ministry of Natural Resources (MNR), Hangzhou 310012, China

**Keywords:** transcriptome, *Patiria pectinifera*, megabenthos, sedimental burial, hypoxic stress, metabolic changes, immune responses

## Abstract

Sediment burial generated by deep-sea mining is usually lethal to echinoderms, which are ecologically important in marine environments. However, their molecular mechanisms responding to sediment burial are still rarely investigated. In this study, *Patiria pectinifera* was investigated for sediment burial research to analyze its gene expression variations by using comparative transcriptomes and to probe into shared molecular mechanisms of echinoderms under sediment burial. During sediment burial experiments, dissolved oxygen continuously decreased, which had a significant impact on *Patiria pectinifera*, which suffered from hypoxic stress. Based on functional annotations of differentially expressed genes (DEGs), its metabolic patterns altered with the upregulated DEGs related to glycolysis and fatty acid degradation and the downregulated ones in the citrate cycle, and its immune responses also varied with the upregulated DEGs of apoptosis and the downregulated ones defending against pathogens. Meanwhile, the peroxisome proliferator-activated receptor signaling pathway and retinoic acid-inducible gene I-like receptor signaling pathway were also upregulated, indicating metabolic and immune changes. Furthermore, combined with functional annotations of twelve echinoderm reference genomes, those DEGs related to lipid metabolism and the immune response were also universally present in the echinoderm genomes. Our study probes into shared molecular mechanisms of echinoderms under sediment burial, which advances our understanding of echinoderms affected by deep-sea mining.

## 1. Introduction

Deep-sea mining aims to access and supply critical metal resources, including barium, cadmium, chromium, cobalt, copper, gold, lead, manganese, molybdenum, nickel, niobium, silver, strontium, titanium, vanadium, zirconium, and zinc, which are becoming increasingly scarce on land, to satisfy demands for social and economic development [1,2,3,4]. Due to environmental disturbances and stresses caused by deep-sea mining, it has potential threats on the deep-sea biome, which covers > 90% of the global biosphere [5]. Therefore, environmental impact assessments need to be conducted and efforts should be made to reduce the negative impacts on the marine environment [6]. The United Nations Convention on the Law of the Sea stipulates that the legal liability for mineral resource development in the ‘Area’ beyond national jurisdiction falls under the purview of the International Seabed Authority (ISA). The ISA has promulgated a draft environmental regulation for deep-sea resource development in the ‘Area’, which requires any state or entity to submit an environmental impact assessment report before conducting deep-sea mining trials [7]. Therefore, environmental impact assessments play a vital role in the feasibility of deep-sea mining.

Deep-sea mining vehicles excavate mineral resources, including polymetallic nodules and cobalt-rich ferromanganese crusts, on the seafloor, which lead to disrupting the original seafloor structure [8]. Deep-sea sediments containing various mineral particles, clays, and biological debris are stirred up and mixed with the surrounding seawater, which generate plume diffusion dispersing across the vast territory of the ocean and significantly influence the marine ecosystem [9,10]. Plume diffusion may bury and smother benthic fauna, resulting in the blockage of suspended feeders and dilution of nutrients during the benthic plume’s deposition [11,12]. Furthermore, deep-sea fauna buried by sediments may be exposed to hypoxic stress [13], which is also lethal to them.

As one of the most noticeable megabenthos on the seafloor, echinoderms (phylum: Echinodermata) consist of more than 7000 species within five classes, including Asteroidea (starfishes), Crinoidea (crinoids), Echinoidea (sea urchins), Holothuroidea (sea cucumbers), and Ophiuroidea (brittle stars) [14]. Echinoderms are widely distributed on the seafloor at all depths from the intertidal flats and shallow-sea sediments to the deepest oceanic trenches [15,16]. Ecologically, echinoderms play an important role in the global carbon cycle, contributing approximately 0.8 Pg of calcium carbonate annually [17]. Deep-sea mining shows considerable negative biological effects on megabenthos, leading to their biodiversity loss [18,19,20]. In situ deep-sea and classical laboratory researches are focused on ecological distribution, behavioral influence, and metabolic variation impacted by sediment burial [21,22,23,24,25]. Recent in situ metabolism determinations for megabenthos revealed no metabolic rate differences between shallow-sea and deep-sea individuals, indicating that experimental researches on shallow-water ones can provide common mechanisms for understanding their responses to sediment burial. Our previous transcriptomic analysis of the brittle star *Ophiothrix exigua* revealed that differentially expressed genes (DEGs) are mostly related to signal pathways, metabolic regulation, and the immune system, causing instability of its internal environment and increasing energy consumption under sediment burial [26]. However, their molecular mechanisms responding to hypoxic stress induced by sediment burial have still rarely been investigated, which hampers our understanding of deep-sea mining’s impacts on echinoderms and megabenthos. In this study, we selected *Patiria pectinifera*, which is commonly used as a model organism in developmental biology [27] for sediment burial research, investigated its gene expression variations by using comparative transcriptomes, and probed into shared molecular mechanisms of echinoderms to hypoxic stress induced by sediment burial, which advances our understanding of echinoderms.

## 2. Results

### 2.1. Survival and Physicochemical Variations at Different Depths of Sediment Burial

During burial by coarse sediments with sediment grain sizes ranging from 63 to 250 μm, *Patiria pectinifera* individuals still survived at burial depths of 2 and 5 cm (Table 1). However, when buried by fine sediments with grain sizes less than 63 μm, their survival rates decreased with the increase in burial depth (Table 1 and Figure 1). In particular, at least one individual demised in the triplicate burial depth of 5 cm using fine sediments.

Physicochemical determinations in fine sediment burial at 5 cm revealed that the dissolved oxygen (DO) decreased significantly (*p* < 0.01), from 68.2 ± 1.4 μmol/L to 4.5 ± 0.3 μmol/L in 24 h. Meanwhile, the other three physicochemical parameters, including salinity, pH, and redox potential, were relatively constant, at approximately 27.6‰ (*w*/*v*), 7.4, and 398.3 mv, respectively (Figure 2).

### 2.2. Transcriptomic Sequence Assembly and Functional Annotations

To compare the transcriptomic responses to sediment burial, three surviving *Patiria pectinifera* individuals that experienced the fine sediment burial depth of 5 cm for 24 h were taken as the experimental group, and another three *Patiria pectinifera* individuals without sediment burial were regarded as the control group. Transcriptomic sequencings generated a total of 27,2386,974 raw reads (average: 45,397,829; median: 44,812,871) with a total of 40,858,046,100 base pairs (average: 6,809,674,350; median: 6,721,930,650). The error rates of those raw reads ranged from 0.22‰ to 0.24‰. After removing low-quality reads, a total of 255,522,878 clean reads with 37,703,265,839 base pairs were retained. Those sequencing reads were deposited into the NCBI Sequence Read Archive database under the accession numbers of SRR31720745 (control group 1), SRR31720744 (control group 2), SRR31720743 (control group 3), SRR31720742 (experimental group 1), SRR31720741 (experimental group 2), and SRR31720740 (experimental group 3). A detailed transcriptomic sequencing summary is listed in Table 2.

De novo transcriptomic assembly revealed that all of the clean reads were assembled into 139,658 transcripts with a total length of 122,084,077 base pairs. Subsequently, those transcripts were processed into 69,028 unigenes with a total length of 53,822,148 base pairs. The longest and average lengths of those unigenes were 17,002 and 780 base pairs, respectively. Functional annotations indicated that 18,973 (27.5%), 14,805 (21.4%), 25,920 (37.6%), 10,667 (15.4%), and 13899 (20.1%) unigenes were found against the Clusters of Orthologous Genes (COG), Gene Ontology (GO), NCBI non-redundant nucleotide, Kyoto Encyclopedia of Genes and Genomes (KEGG), and UniProt databases, respectively (Appendix A).

### 2.3. Annotations of DEGs

Based on the expression level of each transcript calculated according to the reads per kilobase of exon per million mapped reads (RPKM) method, 15,083 unigenes, including 8783 upregulated and 6300 downregulated ones, were identified as DEGs with thresholds of log fold change (|logFC|) ≥ 1 and false discovery rate (FDR) ≤ 0.05 (Figure 3). Among those DEGs, 14,805 and 2742 were annotated against the GO and KEGG databases, respectively.

Functional enrichment analysis against the GO database revealed that DEGs were significantly enriched in three categories (*p* < 0.05), including biological processes (50 level-4 GO terms), cellular components (DEGs, 11 level-4 GO terms), and molecular functions (DEGs, 22 level-4 GO terms). In the biological process category, macromolecule metabolic processes (489 and 482 upregulated and downregulated DEGs), cellular component organization (462 and 471 upregulated and downregulated DEGs), organonitrogen compound metabolic processes (476 and 453 upregulated and downregulated DEGs), cellular macromolecule metabolic processes (418 and 409 upregulated and downregulated DEGs), and protein metabolic processes (379 and 360 upregulated and downregulated DEGs) were the top five level-4 GO terms. In the cellular component category, the IPAF inflammasome complex (65 and 48 upregulated and downregulated DEGs), NLRP3 inflammasome complex (42 and 50 upregulated and downregulated DEGs), inner mitochondrial membrane protein complex (18 and 16 upregulated and downregulated DEGs), oxidoreductase complex (19 and 14 upregulated and downregulated DEGs), respiratory chain complex (15 and 12 upregulated and downregulated DEGs), and mitochondrial respirasome (15 and 12 upregulated and downregulated DEGs) were the top five level-4 GO terms. In the molecular function category, identical protein binding (233 and 208 upregulated and downregulated DEGs), anion binding (183 and 149 upregulated and downregulated DEGs), cation binding (168 and 121 upregulated and downregulated DEGs), protein dimerization activity (133 and 135 upregulated and downregulated DEGs), and nucleotide binding (141 and 120 upregulated and downregulated DEGs) were the top five level-4 GO terms. The top 30 level-4 GO terms annotated by using DEGs are illustrated in Figure 4 and the detailed annotations against the GO database are shown in Appendix A.

Functional enrichment analysis against the KEGG database indicated that DEGs were significantly enriched in five categories (*p* < 0.05), including cellular processes (one layer-3 pathway), environmental information processing (one layer-3 pathway), genetic information processing (three layer-3 pathways), metabolism (14 layer-3 pathways), and organismal systems (five layer-3 pathways). Based on an upregulated/downregulated DEG ratio higher than 2 or less than 0.5, the biosynthesis of unsaturated fatty acids (metabolism, 9 and 4 upregulated and downregulated DEGs), cell adhesion molecules (environmental information processing, 8 and 4 upregulated and downregulated DEGs), fatty acid degradation (metabolism, 17 and 6 upregulated and downregulated DEGs), glycerolipid metabolism (metabolism, 10 and 4 upregulated and downregulated DEGs), glycosphingolipid biosynthesis, i.e., globo and isoglobo series (metabolism, 1 and 7 upregulated and downregulated DEGs), peroxisome proliferator-activated receptor (PPAR) signaling pathway (organismal systems, 15 and 4 upregulated and downregulated DEGs), propanoate metabolism (metabolism, 9 and 3 upregulated and downregulated DEGs), ribosome (genetic information processing, 7 and 54 upregulated and downregulated DEGs), retinoic acid-inducible gene-I-like (RIG-I-like) receptor signaling pathway (organismal systems, 13 and 3 upregulated and downregulated DEGs), RNA polymerase (genetic information processing, 6 and 3 upregulated and downregulated DEGs), and tryptophan metabolism (metabolism, 15 and 6 upregulated and downregulated DEGs) were the most significantly enriched layer-3 pathways. The functionally enriched layer-3 pathways annotated by using DEGs are shown in Figure 5 and the detailed functional enrichments against the KEGG database are listed in Appendix A.

### 2.4. Comparison of DEGs with Other Echinoderms

To elucidate common molecular mechanisms responding to sediment burial among echinoderms, twelve genome sequences of echinoderms, including *Acanthaster planci* (crown-of-thorns starfish), *Amphiura filiformis*, *Anneissia japonica*, *Antedon mediterranea*, *Apostichopus japonicus* (Japanese sea cucumber), *Asterias amurensis*, *Asterias rubens* (European starfish), *Diadema antillarum*, *Lytechinus pictus* (painted urchin) *Lytechinus variegatus* (green sea urchin), *Patiria miniata* (bat star), and *Strongylocentrotus purpuratus* (purple sea urchin), were obtained from the NCBI Reference Genome database (Table 3). Functional annotations of those protein-coding genes revealed that 82.7 to 90.3%, 50.9 to 65.3%, and 58.0 to 65.8% of coding sequences (CDSs) were assigned to the COG, GO, and KEGG databases, respectively (Appendix A).

To elucidate potential molecular mechanisms shared in echinoderms, KEGG annotations based on the most significantly enriched KEGG layer-3 pathways were compared with those of twelve echinoderm genome sequences. Upregulated functional genes in biosynthesis of unsaturated fatty acids, cell adhesion molecules, fatty acid degradation, glycerolipid metabolism, PPAR signaling pathway, propanoate metabolism, RIG-I-like receptor signaling pathway, RNA polymerase, and tryptophan metabolism were almost annotated in the other twelve genome sequences of echinoderms, except for (1) K00452 (3-hydroxyanthranilate 3,4-dioxygenase), absent in the genome of *Apostichopus japonicus*; (2) K03008 (DNA-directed RNA polymerase II subunit RPB11), absent in the genomes of *Acanthaster planci*, *Asterias amurensis*, *Asterias rubens,* and *Patiria miniata*; (3) K03016 (DNA-directed RNA polymerases I, II, and III subunit RPABC3) in the genome of *Lytechinus pictus*; (4) K07296 (adiponectin), absent in the genomes of *Apostichopus japonicus*, *Lytechinus pictus*, *Lytechinus variegatus,* and *Strongylocentrotus purpuratus*; (5) K12298 (bile salt-stimulated lipase), absent in the genomes of *Antedon mediterranea* and *Asterias amurensis*; and (6) K17360 (acyl-coenzyme A thioesterase 7), absent in the genomes of *Apostichopus japonicus*, *Diadema antillarum*, *Lytechinus pictus*, *Lytechinus variegatus,* and *Strongylocentrotus purpuratus*. Downregulated functional genes in glycosphingolipid biosynthesis—globo and isoglobo series—and the ribosome were also encoding the echinoderm reference genomes, other than K01189 (alpha-galactosidase), absent in all genomes, and K02973 (small subunit ribosomal protein S23e), absent in the genome of *Acanthaster planci*. Detailed functional gene comparisons are shown in Appendix A.

## 3. Discussion

### 3.1. Hypoxic Stress as a Major Physicochemical Determinant in the Sediment Burial

Benthos buried by resuspended sediments that are generated by deep-sea mining are subject to respiratory stress due to molecular diffusion of oxygen into the sediment decreasing rapidly, especially for fine-grained sediments [28]. Hypoxic stress aggravates the situation of benthos and eventually leads to their death when they fail to migrate up through the buried sediment [29]. Furthermore, hypoxia induces reactive oxygen species production in the cells, which results in cellular damage [30]. In our sediment burial experiments, DO continuously decreased, which had a significant impact on *Patiria pectinifera*, which suffered from hypoxic stress. In the natural environment, oxygen availability is one of the pivotal factors affecting benthic communities and distributions [31]. Therefore, extended periods of hypoxia can be lethal to benthos when they are buried by sediments.

### 3.2. Metabolic Responses to Hypoxic Stress

Hypoxic stress is considered as one significant factor altering metabolic patterns [32]. (1) Glycolysis plays a key role in anaerobic metabolism when exposed to hypoxic stress [33]. In our study, glycolysis-related DEGs, including *ALDH* (aldehyde dehydrogenase) and *pdhD* (dihydrolipoyl dehydrogenase), were upregulated, with fold-changes of 9 to 55 when exposed to sediment burial. (2) Hypoxic stress interrupts the further oxidation of glucose, which decreases the levels of the citric acid cycle [34]. In our study, citrate cycle-related DEGs, including *acnA* (aconitate hydratase), *fumA* (fumarate hydratase), *icd* (isocitrate dehydrogenase), *LSC1* (succinyl-CoA synthetase alpha subunit), *pdhD* (dihydrolipoyl dehydrogenase), and *SDH1* (succinate dehydrogenase (ubiquinone) flavoprotein subunit), were downregulated with fold changes of 16 to 4312 under sediment burial. (3) Under hypoxic stress, glucose is converted into pyruvate, which is subsequently metabolized to lactate or succinate as the end product, which leads to the accumulation of lactate and succinate [35,36]. In our study, pyruvate metabolism-related DEGs, including *ALDH* (aldehyde dehydrogenase), *gloB* (hydroxyacylglutathione hydrolase), *GRHPR* (glyoxylate/hydroxypyruvate reductase), *lpd* (dihydrolipoyl dehydrogenase), and *MDH2* (malate dehydrogenase), were upregulated with fold changes of 9 to 3278 with decreasing dissolved oxygen. (4) Lipid are utilized as an alternative carbon source in energy supply through other metabolic pathways, such as fatty acid degradation [37]. In our study, fatty degradation-related DEGs, including *ACAA1* (acetyl-CoA acyltransferase 1), *ACOX1* (acyl-CoA oxidase), *ALDH* (aldehyde dehydrogenase), *CYP2U1* (long-chain fatty acid omega-monooxygenase), *ECI1* (delta3-delta2-enoyl-CoA isomerase), *EHHADH* (enoyl-CoA hydratase), *fadD* (long-chain-fatty-acid-CoA ligase), *gcdH* (glutaryl-CoA dehydrogenase), and *HADHB* (acetyl-CoA acyltransferase), were upregulated with fold changes of 15 to 27,478 when exposed to the sediment burial. (5) Hypoxic stress often enhances the production of reactive oxygen species and leads to a significant increase in glutathione which is a major antioxidant-removing reactive oxygen species [38]. In our study, glutathione synthesis- and ROS removal-related DEGs, including *gpx* (glutathione peroxidase) and *gshB* (glutathione synthase), were upregulated, with fold-changes of 26 to 1319, while glutathione degradation-related DEGs, including *gst* (glutathione S-transferase), were downregulated, with fold changes of 12 to 211. Furthermore, KEGG functional enrichment analysis and annotations of twelve echinoderm reference genomes revealed that the majority of upregulated metabolic pathways in our study were related to lipid metabolism, including the biosynthesis of unsaturated fatty acids, fatty acid degradation, and glycerolipid metabolism, indicating that echinoderms tend to utilize fatty acid as an energy source when they suffer from hypoxic stress induced by sediment burial.

### 3.3. Immune Responses to Hypoxic Stress

Acute hypoxic stress induces various immune responses, including apoptosis [32]. (1) Hypoxic stress triggers apoptosis [39], which results in damage to the tissue structures of *Patiria pectinifera*. In our study, apoptosis-related DEGs, including *ACTB_G1* (actin beta/gamma 1), *CASP8* (caspase 8), *CASP9* (caspase 9), *CTSB* (cathepsin B), *CTSC* (cathepsin C), *CTSL* (cathepsin L), *EIF2AK3* (eukaryotic translation initiation factor 2-alpha kinase 3), *MAP3K5* (mitogen-activated protein kinase kinase kinase 5), *PDPK1* (3-phosphoinositide dependent protein kinase-1), *TUBA* (tubulin alpha), and *TRAF2* (TNF receptor-associated factor 2), were upregulated, with fold changes of 11 to 18,442. (2) Hypoxic stress decreases resistance to pathogen infection by decreasing phagocytosis rates [40]. In our study, phagocytosis-related DEGs, including *ARPC3* (actin-related protein 2/3 complex, subunit 3), *crk* (C-crk adapter molecule), and WASF3 (WAS protein family, member 3), were downregulated with fold changes of 2537 to 5034. Based on those variations, hypoxic stress induces apoptosis and weakens the response of the immune system, which threaten the survival of echinoderms under sediment burial.

### 3.4. Signaling Pathway Responses to Hypoxic Stress

Signaling pathways are vital to regulating the stress response to oxidative stress induced by hypoxia in aquatic animals [41,42,43]. (1) Hypoxia-inducible factors (HIFs) act as a pivotal regulator responding to decreases in available oxygen, and they are usually highly expressed to cope with hypoxic stress [44]. In our study, HIF-1 signaling pathway-related DEGs, including *CUL2* (cullin 2), *ELOB* (elongin-B), and *PFKFB3* (6-phosphofructo-2-kinase), were upregulated with fold changes of 13 to 2847. (2) Excessively generated reactive oxygen species activate mitogen-activated protein kinase (MAPK) signaling pathway proteins, eventually activating the HIF-1α protein [45]. During sediment burial, MAPK pathway-related genes, including *azk* (sterile alpha motif and leucine zipper containing kinase), *FGFR3* (fibroblast growth factor receptor 3), *FGFR4* (fibroblast growth factor receptor 4), *MAP3K5* (mitogen-activated protein kinase 5), *met* (proto-oncogene tyrosine-protein kinase), *RAC1* (Ras-related C3 botulinum toxin substrate 1), *ret* (proto-oncogene tyrosine-protein kinase), *TRAF2* (TNF receptor-associated factor 2), and *TRAF6* (TNF receptor-associated factor 6), were upregulated, with fold changes of 10 to 21,058. (3) PPAR signaling pathways are responsible for regulating gene transcriptions in lipid metabolism and immune responses of aquatic invertebrates [46,47,48]. In our study, PPAR signaling pathway-related DEGs comprising *ACAA1* (acetyl-CoA acyltransferase 1), *ACOX1* (acyl-CoA oxidase), *ADIPOQ* (adiponectin), *DBI* (diazepam-binding inhibitor), *EHHADH* (enoyl-CoA hydratase), *FABP3* (fatty acid-binding protein 3, muscle, and heart), *fadD* (long-chain fatty-acid CoA ligase), *FADS2* (acyl-CoA 6-desaturase), *PDPK1* (3-phosphoinositide dependent protein kinase-1), *glpK* (glycerol kinase), and *SLC27A6* (solute carrier family 27 (fatty acid transporter), member 6) were upregulated with fold changes of 13 to 27,478. (4) The RIG-I-like receptor signaling pathway plays a vital role in detecting viral pathogens and generating innate immune responses [49,50,51]. In our study, RIG-I-like receptor signaling pathway-related DEGs, including *ATG5* (autophagy-related protein 5), *CASP8* (caspase 8), *CYLD* (ubiquitin carboxyl-terminal hydrolase), OTUD5 (OTU domain-containing protein 5), *TKFC* (triose/dihydroxyacetone kinase), *TRAF2* (TNF receptor-associated factor 2), *TRAF3* (TNF receptor-associated factor 3), and *TRAF6* (TNF receptor-associated factor 6) were upregulated, with fold changes of 38 to 11,254. Moreover, most of the DEGs in the PPAR signaling pathway and RIG-I-like receptor signaling pathway were also annotated in the twelve echinoderm reference genomes, indicating that lipid metabolism and immune response changes are major mechanisms responding to hypoxic stress generated by sediment burial.

## 4. Materials and Methods

### 4.1. Preparison of Deep-Sea Sediments and Rearing of Patiria pectinifera

Deep-sea sediments and echinoderm individuals for sediment burial were prepared and processed as described by Wang et al. [26]. Sediments were collected from the polymetallic nodule area in the East Pacific Ocean and then cleaned and sieved using freshwater in a vibrating sieve machine, with sieve meshes of 250 μm and 63 μm aperture sizes. Subsequently, those sediments were employed to be separated and stored as two distinct particle-size fractions, including fine sediments (particle size smaller than 63 μm) and coarse sediments (particle size ranging between 63 and 250 μm), and were prepared for use in sediment burial experiments. Thereafter, the separated sediment fractions were subjected to immersion in artificial seawater with a salinity of 31‰ (*w*/*v*). Blue bat star *Patiria pectinifera* individuals were collected from the coast of Qingdao, China (120.38° E, 36.06° N) at a depth of 6 m in 2023 and were temporarily kept in the laboratory aquarium for five days.

### 4.2. Sediment Burial

The size of the aquarium used in this experiment was 100 cm × 50 cm × 50 cm. The sediment burial thickness was set at 5 cm, and the burial time was set at 24 h, as described by Hendrick et al. [23]. The sediment thicknesses were set to 2 cm and 5 cm, as 5 cm is the median expected increase in sediment levels during marine aggregate dredging activity [26]. The experiment consisted of a sediment burial group and a control group, both containing seven individuals, with three replicates for each group. *Patiria pectinifera* in the control group were placed in the same type of aquarium with no sediment burial. After the burial time, tissue samples taken from surviving *Patiria pectinifera* were instantly transferred in liquid nitrogen for short-term freezing and then preserved in an ultra-low-temperature refrigerator at −80 °C for long-term storage.

### 4.3. RNA Extraction, Library Preparation and Illumina Hiseq Sequencing

Total RNA was extracted from the tissue by using TRIzol^®^ Reagent according to the manufacturer’s instructions (Invitrogen, Waltham, MA, USA), with genomic DNA removed by using DNase I (TaKaRa Biomedical Technology Co., Ltd., Beijing, China). High-quality RNA samples (OD_260/280_ = 1.8~2.2, OD_260/230_ ≥ 2.0, RIN ≥ 6.5, 28S:18S ≥ 1.0, >10 μg) were applied to construct the sequencing library, following RNA quality determination using a 2100 Bioanalyzer (Agilent Technologies, Santa Clara, CA, USA) and quantification using the ND-2000 (NanoDrop Technologies, Thermo Fisher Scientific Inc., Waltham, MA, USA).

RNA-seq transcriptome libraries were prepared by using the TruSeqTM RNA sample preparation Kit from Illumina (San Diego, CA, USA). In detail, cDNA synthesis, end repair, A-base addition, and adaptor ligation were carried out according to Illumina’s protocol, after messenger RNA was isolated with polyA selection using oligo(dT) beads and fragmented by fragmentation buffer. Sequencing libraries were then size-selected for cDNA target fragments of 200–300 bp on 2% (*w*/*v*) low-range ultra agarose followed by PCR amplification using Phusion DNA polymerase (New England Biolabs, Ipswich, MA, USA). After quantification by TBS380 picogreen (Thermo Fisher Scientific, Waltham, MA, USA), paired-end library sequencing with a length of 150 bp was performed by Illumina NovaSeq 6000 sequencing (Shanghai BIOZERON Co., Ltd., Shanghai, China).

### 4.4. De Novo Assembly and Functional Annotations

Raw paired-end reads were trimmed and quality-controlled by Trimmomatic v0.36 with parameters (SLIDINGWINDOW:4:15 MINLEN:75) [52] for quality-control. Subsequently, clean reads were assembled de novo into transcripts using Trinity v2.15.1 [53]. All the assembled transcripts were searched against the NCBI protein nonredundant database using BLASTX to identify the proteins with typical cut-off E-values less than 1.0 × 10^−5^ [54]. GO annotations of unique assembled transcripts for describing biological processes, molecular functions, and cellular components were obtained by using the BLAST2GO v6.0 program [55]. KEGG annotations of unique assembled transcripts were conducted by using eggNOG-mapper v2.1.8 with protein alignments using diamond [56,57]. In addition, twelve genome sequences of echinoderms were obtained from the NCBI Reference Genome database and their KEGG annotations were also performed as described above.

### 4.5. Differential Expression Analysis and Functional Enrichment Analysis

To identify DEGs between the two sediment burial groups and the control group, the expression level of each transcript was calculated according to the RPKM method. Gene quantifications and isoform abundance calculations were carried out by using RNA-Seq in Expectation-Maximization software v1.3.3 [58]. Differential expression analysis was performed using the R statistical package software edgeR v3.18.1 [59]. In addition, functional enrichment analysis, including GO and KEGG, were performed to identify which DEGs were significantly enriched in GO terms and metabolic pathways at Bonferroni-corrected *p* ≤ 0.05 compared with the whole-transcriptome background. GO functional enrichment and KEGG pathway analysis were carried out by Goatools v1.4.5 [60] and KOBAS v2.0 [61].

## 5. Conclusions

Sediment burial generated by deep-sea mining is usually lethal to megabenthos, which are ecologically important in marine environments. Our study demonstrates that acute hypoxic stress occurred during our simulated sediment burial of *Patiria pectinifera*, which exhibited decreased survival when buried by fine sediments. Based on GO and KEGG annotations of DEGs between the sediment burial group and the control group of *Patiria pectinifera*, its metabolic patterns changed with the upregulated DEGs related to glycolysis and fatty acid degradation and the downregulated ones in the citrate cycle, and its immune responses also varied with the upregulated DEGs of apoptosis and the downregulated ones defending against pathogens. Furthermore, the PPAR signaling pathway and RIG-I-like receptor signaling pathway were also upregulated, indicating metabolic and immune changes. Functional enrichment analysis and KEGG annotations of echinoderm reference genomes revealed that those DEGs related to lipid metabolism and immune response are also universally present in echinoderm genomes. Our study demonstrates that echinoderms attempt to survive by changing metabolic patterns when exposed to sediment burial, while increasing apoptosis and decreased immune responses to pathogens may cause their deaths. While the response of echinoderms to sediment burial and their survival rates vary over time, a time-series determination of transcriptomic variations can provide a more comprehensive understanding of their molecular mechanism responding to external stress in the future. Our study investigates gene expression variations by using comparative transcriptomes and probes into shared molecular mechanisms of echinoderms to hypoxic stress induced by sediment burial, which advances our understanding of echinoderms affected by deep-sea mining.

## Figures and Tables

**Figure 1 ijms-26-05208-f001:**
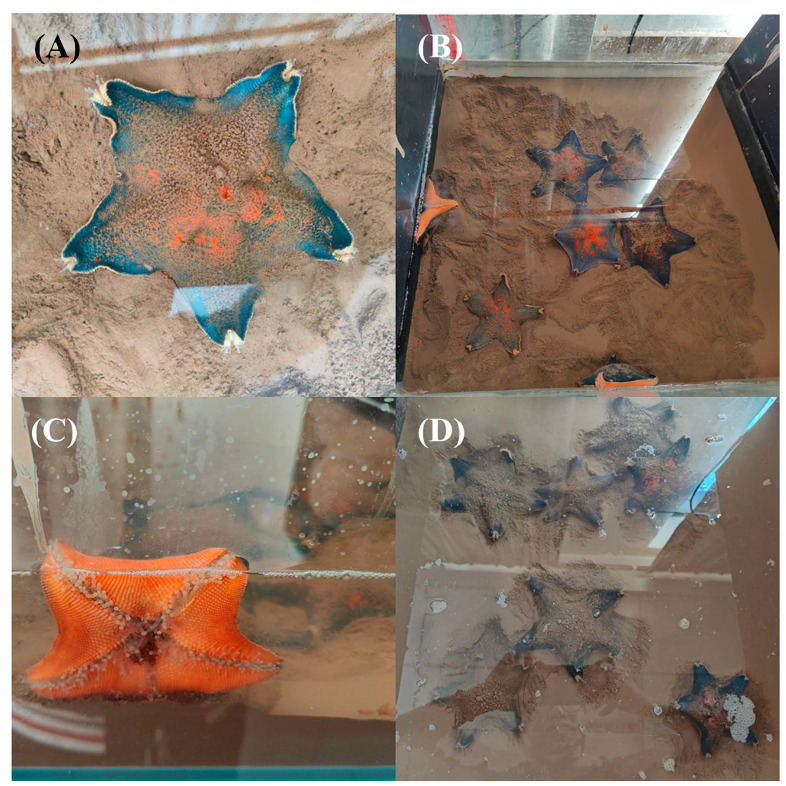
*Patiria pectinifera* individuals were buried with fine sediments at burial depths of 2 cm (**A**,**B**) and 5 cm (**C**,**D**).

**Figure 2 ijms-26-05208-f002:**
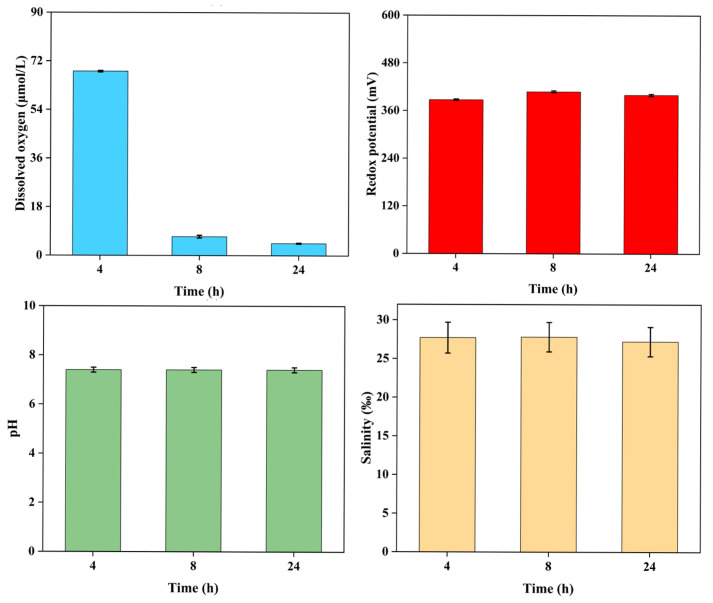
Physicochemical variations under fine sediment burial of *Patiria pectinifera* with a depth of 5 cm for 24 h. Blue, yellow, green, and red column charts indicate dissolved oxygen, redox potential, pH, and salinity, respectively.

**Figure 3 ijms-26-05208-f003:**
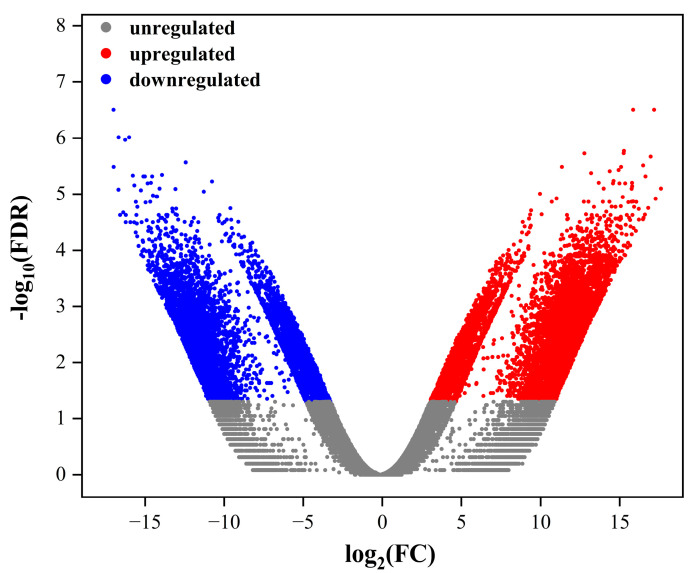
Volcano plot of DEGs between the experimental group and sediment burial group of *Patiria pectinifera*. Red, blue, and grey indicate upregulated, downregulated, and unregulated DEGs, respectively.

**Figure 4 ijms-26-05208-f004:**
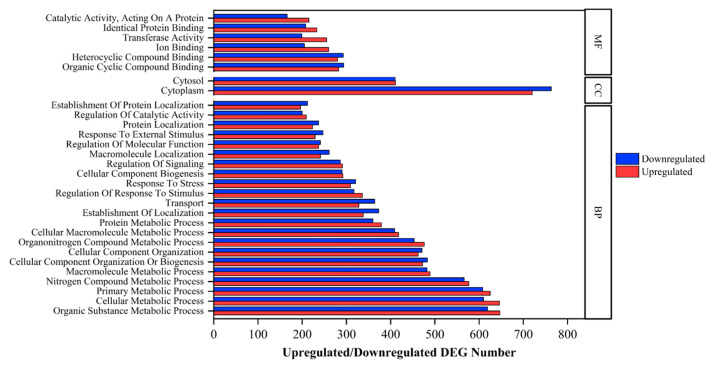
Top 30 GO level-4 term enrichment analysis (*p* < 0.05) with its regulated DEG numbers. BP, CC, and MF indicate biological process, cellular component, and molecular function, respectively. Red and blue indicate upregulated and downregulated DEG, respectively.

**Figure 5 ijms-26-05208-f005:**
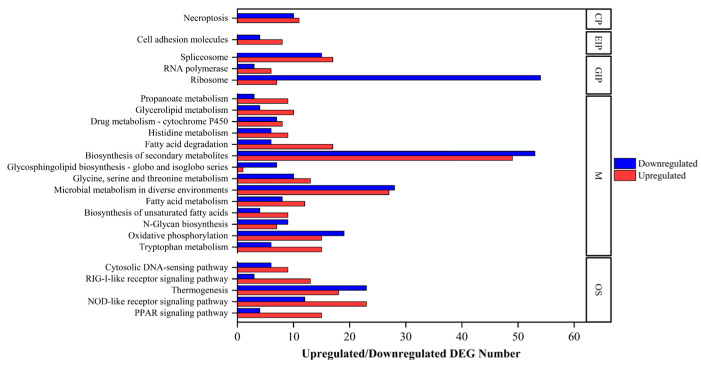
KEGG layer-3 pathway enrichment analysis (*p* < 0.05) with its regulated DEG numbers. CP, EIP, GIP, M, and OS indicate cellular processes, environmental information processing, genetic information processing, metabolism, and organismal systems, respectively. Red and blue indicate upregulated and downregulated DEGs, respectively.

**Table 1 ijms-26-05208-t001:** Survival rates of *Patiria pectinifera* individuals under sediment burial with different grain sizes after 24 h. Seven individuals were selected in one aquarium, and those experiments were performed in triplicate.

Sediment Grain Size (μm)	Burial Depth (cm)	Survival Rates (%)
<63	2	95.2
	5	81.0
63 to 250	2	100.0
	5	100.0

**Table 2 ijms-26-05208-t002:** Summary of transcriptome sequencing datasets of *Patiria pectinifera* with or without sediment burial at 5 cm.

Sample	Raw Reads	Raw Bases(bp)	Clean Reads	Clean Bases(bp)	Valid Bases(%)	GC Content(%)
Experimental group 1	4,7078,864	7,061,829,600	44,362,144	6,577,106,998	93.1	45.1
Experimental group 2	48,983,870	7,347,580,500	46,163,352	6,844,046,669	93.1	45.1
Experimental group 3	44,980,718	6,747,107,700	42,781,734	6,357,259,233	94.2	45.2
Control group 1	43,523,738	6,528,560,700	40,288,662	5,897,539,644	90.3	46.0
Control group 2	43,174,760	6,476,214,000	39,964,578	5,850,154,113	90.3	46.0
Control group 3	44,645,024	6,696,753,600	41,962,408	6,177,159,182	92.2	46.3

**Table 3 ijms-26-05208-t003:** Genomic information of twelve echinoderm genomes obtained from NCBI Reference Genome database.

Class	Species	Accession Number	Size (Mbp)	GC Content (%)	Transcripts	CDSs
Asteroidea	*Acanthaster planci*	GCF_001949145.1	383.8	41.5	36,221	33,201
	*Asterias amurensis*	GCF_032118995.1	491.5	39.0	30,608	26,569
	*Asterias rubens*	GCF_902459465.1	417.6	39.0	28,238	24,049
	*Patiria miniata*	GCF_015706575.1	608.3	40.5	40,283	35,403
Crinoidea	*Anneissia japonica*	GCF_011630105.1	589.6	34.5	38,110	32,774
	*Antedon mediterranea*	GCF_964355755.1	354.5	33.5	33,480	25,416
Echinoidea	*Diadema antillarum*	GCF_040938485.1	1800.0	38.5	39,579	36,155
	*Lytechinus pictus*	GCF_037042905.1	811.7	36.5	37,286	29,875
	*Lytechinus variegatus*	GCF_018143015.1	869.6	36.5	38,196	33,669
	*Strongylocentrotus purpuratus*	GCF_000002235.5	921.8	37.5	45,168	38,439
Holothuroidea	*Apostichopus japonicus*	GCF_037975245.1	671.6	37.5	57,784	45,056
Ophiuroidea	*Amphiura filiformis*	GCF_039555335.1	1600.0	36.5	46,206	39,055

## Data Availability

Transcriptomic reads are deposited in the NCBI Sequence Read Archive database under the accession numbers of SRR31720745 (control group 1), SRR31720744 (control group 2), SRR31720743 (control group 3), SRR31720742 (experimental group 1), SRR31720741 (experimental group 2), and SRR31720740 (experimental group 3).

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
