# Peer review of "Transcriptomic Responses of Blue Bat Star *Patiria pectinifera* to Sediment Burial"

_ijms, 2025, doi:10.3390/ijms26115208_

Round 1
Reviewer 1 Report
Comments and Suggestions for Authors
This study investigates the transcriptomic responses of the blue bat star Patiria pectinifera to sediment burial, revealing significant metabolic shifts,such as upregulated glycolysis and fatty acid degradation, downregulated citrate cycle, and immune alterations ,apoptosis activation, pathogen defense suppression. Comparative genomic analysis across 12 echinoderm species further identifies conserved molecular mechanisms underlying these responses. The work provides critical insights into the molecular adaptations of echinoderms to sediment burial stress, advancing our understanding of ecological impacts caused by deep-sea mining.
Required Revisions:
1. A photograph of Patiria pectinifera should be included to aid reader comprehension and validate experimental procedures.
2. The study does not address potential confounding effects of sediment composition (e.g., heavy metals, organic pollutants). Chemical profiling of sediments is essential to isolate burial-induced stress from contaminant-driven transcriptional changes, especially given that mining-derived sediments vary widely in toxicity.
Author Response
Reviewer#1
This study investigates the transcriptomic responses of the blue bat star Patiria pectinifera to sediment burial, revealing significant metabolic shifts, such as upregulated glycolysis and fatty acid degradation, downregulated citrate cycle, and immune alterations, apoptosis activation, pathogen defense suppression. Comparative genomic analysis across 12 echinoderm species further identifies conserved molecular mechanisms underlying these responses. The work provides critical insights into the molecular adaptations of echinoderms to sediment burial stress, advancing our understanding of ecological impacts caused by deep-sea mining.
Reply: Thanks for your positive comments on our manuscript. We have revised this manuscript according to your insightful recommendations.
Required Revisions:
- A photograph of Patiria pectinifera should be included to aid reader comprehension and validate experimental procedures.
Reply: As suggested, we have added four photographs including Patiria pectinifera individuals buried by fine sediments with burial depths of 2 cm (A and B) and 5 cm (C and D) as Figure 1.
- The study does not address potential confounding effects of sediment composition (e.g., heavy metals, organic pollutants). Chemical profiling of sediments is essential to isolate burial-induced stress from contaminant-driven transcriptional changes, especially given that mining-derived sediments vary widely in toxicity.
Reply: Thanks for this insightful recommendation. As pointed out by you, heavy metal and organic pollutant pollutions commonly arise during the deep-sea mining. While those pollutions are considered as long-term and far-field impacts to key megabenthos (doi: 10.3389/fmars.2017.00368; doi: 10.1016/j.proeps.2015.06.026). And our study is focused on the sediment burial, which is usually an acute stress to megabenthos (doi: 10.1038/s41467-023-43023-6). Moreover, in situ and simulation experiments about those pollutions on megabenthos are carried out by adding reagents, rather than using sediments in situ (doi: 10.1016/j.scitotenv.2024.173184; doi: 10.1016/j.aquatox.2021.105845). In our study, sediments used in this study were prepared by cleaning using freshwater as described in line 340, in order to remove possible impacts caused by those compounds. Thus, those compound concentrations were not determined in this study.

Reviewer 2 Report
Comments and Suggestions for Authors
The topic is interesting and in general the paper is well presented, but I think there are areas for improvement.
There are many abbreviations used, a table of these is placed at the end of the paper but this does not cover all the abbreviations used (some are referred to in figure captions or elsewhere in the text). It would be best to have a more comprehensive table and place this earlier in the paper (between introduction and results) so that the reader is aware of it early on.
Abbreviations used in the abstract should all be defined in the abstract, otherwise it does not make sense when seen separately.
Figure 2 is not referred to in the text. I assume the authors intended this to be mentioned somewhere around line 134.
The term 'low oxygen stress' could be ambiguous in English: it could mean 'low-oxygen stress' or 'low oxygen-stress'. 'Hypoxic stress' would be better.
The discussion repeats the names of a very large number of genes and is actually not very useful. It would be better to summarise the metabolic pathways referred to and explain or form hypotheses about the biological relevance of these pathways. Also explain why you used 24 hr burial time and not a range of times - have other studies used a range of times, and if so what did they find?
Please summarise what new information or new approach your study adds to the existing literature.
Author Response
Reviewer#2
The topic is interesting and in general the paper is well presented, but I think there are areas for improvement.
Reply: Thanks for your positive comments on our manuscript. We have revised this manuscript according to your insightful recommendations.
There are many abbreviations used, a table of these is placed at the end of the paper but this does not cover all the abbreviations used (some are referred to in figure captions or elsewhere in the text). It would be best to have a more comprehensive table and place this earlier in the paper (between introduction and results) so that the reader is aware of it early on.
Reply: As suggested, we have added abbreviations including BP, CC, CP, EIP, GIP, M, MF and OS in the Abbreviation list (page 13). According to the manuscript template of International Journal of Molecular Science indicating that the abbreviation list is located before the References, we still put the table at the previous position.
Abbreviations used in the abstract should all be defined in the abstract, otherwise it does not make sense when seen separately.
Reply: As suggested, we have added full names of PPAR and RIG in lines 23-24.
Figure 2 is not referred to in the text. I assume the authors intended this to be mentioned somewhere around line 134.
Reply: Apologize for our mistake. We have added this figure citation in line 136.
The term 'low oxygen stress' could be ambiguous in English: it could mean 'low-oxygen stress' or 'low oxygen-stress'. 'Hypoxic stress' would be better.
Reply: As suggested, we have corrected the term to ‘hypoxic stress’ (lines 19, 31-32, 58, 76, 82, 233, 236, 241, 245-248, 250, 256, 269, 280-284, 290, 294, 298, 302, 330, 405, 423).
The discussion repeats the names of a very large number of genes and is actually not very useful. It would be better to summarise the metabolic pathways referred to and explain or form hypotheses about the biological relevance of these pathways. Also explain why you used 24 hr burial time and not a range of times - have other studies used a range of times, and if so what did they find?
Reply: As suggested, we have deleted unigene IDs from the manuscript (lines 247-274, 284-293, 300-327), gene name interpretations are retained for readers to understand those details easily. And summaries of metabolic changes are listed at the end of each subsection in the Discussion (lines 275-280, 294-296, 327-330).
Our sediment burial experiments indicated that a minor of Patiria pectinifera individuals demised and a major of them still survived after the burial of 24 hours, which revealed that survival individuals was reacting against this stress, indicating that transcriptomic changes could be considered as its molecular mechanisms responding to sediment burial. We used the key word “transcriptome, sediment burial, benthos” to search the top 100 published scientific article by Google Scholoar, most of studies were focused on microbes, and those megabenthos transcriptomic studies including Corbicula fluminea (doi: 10.1016/j.scitotenv.2020.138821), Goniastrea pectinata (doi: 10.1111/mec.16263), Mya truncata (doi: 10.1007/s12192-018-0910-5), Mycedium elephantotus (doi: 10.1111/mec.16263), Ophiothrix exigua (doi: 10.1007/s10750-023-05271-x), Ruditapes philippinarum (doi: 10.1016/j.fsi.2018.11.043; doi: 10.3390/biology13110870; doi: 10.1186/s12864-020-6734-6; doi: 10.3389/fmars.2022.845768; doi: 10.1016/j.jtherbio.2023.103776), Sinonovacula constricta (doi: 10.1016/j.scitotenv.2019.136280) were mainly designed to collect tissue samples at a certain moment. While we still appreciate your insightful recommendation, arousing our interest to track time-series transcriptomic changes of echinoderms under different stresses.
Please summarise what new information or new approach your study adds to the existing literature.
Reply: Based on the Google Scholar searches mentioned in the previous comments, only one echinoderm transcriptomic study responding to sediment burial was carried out previously (doi: 10.1007/s10750-023-05271-x) indicating that the most enriched GO terms of differentially expressed genes act on L-amino acid peptides, peptidase activity, transport vesicle, phagocytic vesicle, zymogen granule membrane, basement membrane, vesicle lumen, secretory granule and induction of programmed cell death, while the results of the KEGG pathway enrichment analysis of differentially expressed genes showed that sediment burial will cause different levels of impacts on brittle stars, including changes in p53 signaling pathway, Hippo signaling pathway, MAPK signaling pathway and metabolic pathways. Compared with this study, we found that the key physicochemical factor during the sediment burial was the decrease of dissolved oxygen, and the key transcriptomic changes were related to the upregulations of glycolysis, fatty acid degradation, apoptosis, PPAR signaling pathway and RIG-I-like receptor signaling pathway and the downregulations of citrate cycle, immune responses defending against pathogens. Furthermore, we carried out echinoderm reference genome mapping to detect those differentially expressed genes related to lipid metabolism and immune response are also universally present. Those new insights were shown in lines 403-414.
Reviewer 3 Report
Comments and Suggestions for Authors
General comments:
Comments for review of the manuscript entitled: “ Transcriptomic responses of blue bat star Patiria pectinifera to sediment burial”.
The topics covered in the manuscript about gene expression variations by using comparative transcriptomes and probes into shared molecular mechanisms of Patiria pectinifera to low oxygen stress induced by sediment burial are interesting and use appropriate methods.
The manuscript is very extensive, but has not lost any of its structure due to its comprehensiveness. The experiments were well created and to confirm the results of this research. The analyses were carried out properly with obvious experience from the research group regarding the topic.
Line by Line Comments
Title
My suggestion is to insert species Latin name and author name.
Abstract
Line 14 I suggest: „Patiria pectinifera was investigated…“ instead „we investigated…“.
Keywords
Good choice of keywords, which are almost not reported in the title and will improve the soundness of this manuscript.
Introduction
The introduction is well written, understandable and in a logical order. The problem of deep-sea mining and the role of Echinoderms in this environmental challenge are clearly presented. I think that highlighting the legal regulation is very positive and necessary.
Results
The results are written in a logical sequence and well explained.
Line 97-101 I suggest moving these lines and instead of four graphs for parameters redox, salinity, pH and oxygen, only display data for oxygen. Namely, it is to be expected that the pH is constant considering the property of the sea as a buffer.
Discussion
Lines 236-242 This part seems like an introduction to me, and here it only serves as a better explanation. Discussion is in lines 233 – 236. I suggest the authors reorganize this section.
Line 321 Number 1…is that mistake?
Materials and methods
Line 374 „…previously?“ Do you mean like in the reference number 26? It seems better to me the short description or write according…
Conclusions
I suggest the authors to emphasize the limitations of the research as well as the possibilities of future research.
References
The references list is relevant and has mostly recent publications.
Author Response
Reviewer#3
General comments:
Comments for review of the manuscript entitled: “ Transcriptomic responses of blue bat star Patiria pectinifera to sediment burial”.
The topics covered in the manuscript about gene expression variations by using comparative transcriptomes and probes into shared molecular mechanisms of Patiria pectinifera to low oxygen stress induced by sediment burial are interesting and use appropriate methods.
The manuscript is very extensive, but has not lost any of its structure due to its comprehensiveness. The experiments were well created and to confirm the results of this research. The analyses were carried out properly with obvious experience from the research group regarding the topic.
Reply: Thanks for your positive comments on our manuscript. We have revised this manuscript according to your insightful recommendations.
Line by Line Comments
Title
My suggestion is to insert species Latin name and author name.
Reply: The Latin name “Patiria pectinifera” was line 2 and author names was in line 4.
Abstract
Line 14 I suggest: “Patiria pectinifera was investigated…” instead “we investigated…”.
Reply: Changed as suggested (line 15).
Keywords
Good choice of keywords, which are almost not reported in the title and will improve the soundness of this manuscript.
Reply: Thanks for your positive comments.
Introduction
The introduction is well written, understandable and in a logical order. The problem of deep-sea mining and the role of Echinoderms in this environmental challenge are clearly presented. I think that highlighting the legal regulation is very positive and necessary.
Reply: Thanks for your positive comments.
Results
The results are written in a logical sequence and well explained.
Reply: Thanks for your positive comments.
Line 97-101 I suggest moving these lines and instead of four graphs for parameters redox, salinity, pH and oxygen, only display data for oxygen. Namely, it is to be expected that the pH is constant considering the property of the sea as a buffer.
Reply: Thanks for your suggestion. While other three physicochemical factors were shown to indicate that Patiria pectinifera individuals were in the stable environment, except for the changes of dissolved oxygen.
Discussion
Lines 236-242 This part seems like an introduction to me, and here it only serves as a better explanation. Discussion is in lines 233 – 236. I suggest the authors reorganize this section.
Reply: Thanks for your recommendation. We reorganized this section by moving discussion downward (lines 234-242).
Line 321 Number 1…is that mistake?
Reply: Same as previous discussion in subsections of “3.2 Metabolic responses to hypoxic stress” and “3.3 Immune responses to hypoxic stress”, those numbers were listed to separate different signaling pathways including HIF‐1 signaling pathway, MAPK signaling pathway, PPAR signaling pathway, RIG‐I‐like signaling pathway.
Materials and methods
Line 374 „…previously?“ Do you mean like in the reference number 26? It seems better to me the short description or write according…
Reply: Changed as suggested (line 335).
Conclusions
I suggest the authors to emphasize the limitations of the research as well as the possibilities of future research.
Reply: Thanks for your insightful recommendation. As pointed out by reviewer#2, a time-series investigations can provide a more comprehensive understandings of their molecular mechanism responding to external stress in the future. And those descriptions have been added in lines 418-421.
References
The references list is relevant and has mostly recent publications.
Reply: Thanks for your positive comments.

Round 2
Reviewer 1 Report
Comments and Suggestions for Authors
Accept
Reviewer 2 Report
Comments and Suggestions for Authors
I thank the authors for their responses to my comments and their modifications to the paper. I think the paper is now suitable for publication.